# ELICITING HUMAN PREFERENCES WITH LANGUAGE MODELS

## ABSTRACT

As language models (LMs) become more capable, they are being applied to increasingly complex and user-specific tasks. LMs can be directed to perform target tasks by using labeled examples or natural language prompts, which may include general, free-form task descriptions. But selecting examples or writing prompts for an LM can be challenging—especially in tasks that involve unusual edge cases, demand precise articulation of nebulous preferences, or require an accurate mental model of LMs themselves. We propose to use *LMs themselves* to guide the task specification process. In this paper, we introduce **generative active task elicitation (GATE)**: a learning framework in which models elicit and infer intended behavior through free-form, language-based interaction with users. We study GATE in three domains: email validation, content recommendation, and moral reasoning. In preregistered experiments, we show that LMs prompted to perform GATE (e.g., by generating open-ended questions or synthesizing informative edge cases) elicit responses that are often more informative than user-written prompts or labels. Users report that interactive task elicitation requires less effort than prompting or example labeling and surfaces novel considerations not initially anticipated by users. Our findings suggest that LM-driven elicitation can be a powerful tool for aligning models to complex human preferences and values.

## 1 INTRODUCTION

The complexity of human preferences makes them challenging to encode in machine learning systems. Consider the problem of designing a recommendation system for songs or websites: first, system builders must develop a formal model of the potential factors influencing user preferences; second, users must describe their preferences in a format that a learning algorithm can use to make future recommendations. Each of these steps requires mental effort and continual refinement by users and system builders. Until recently, the dominant approach in machine learning has specified preferences using *examples*: users first label a dataset with examples of the desired model behavior, then train a machine learning model on this dataset. This strategy has seen widespread use across diverse tasks, including image classification and question answering. (Krizhevsky et al., 2012; Devlin et al., 2019). In more recent years, this paradigm has changed with the advent of *instruction following* methods (Brown et al., 2020a): by pre-training langauge models (LMs) on large-scale text corpora, it is possible to infer desired behaviors conditioned only on natural language task specifications, in tasks as diverse as code generation and text summarization.

However, this progress has also accentuated the challenges described above: complex behaviors require an increasing amount of *prompt engineering* or *dataset design* to overcome the imprecision of natural language and prevent models from misunderstanding or misgeneralizing from spurious features of prompts or examples. For example, a user who says they enjoy reading tennis articles could either be interested in the competitive tennis circuit or in improving their own serve. A few user-provided examples of tennis-related articles might fail to specify whether the user is interested in broader tennis content, such as tennis-themed satire. These challenges of *task ambiguity* (Finn et al., 2018; Tamkin et al., 2022a) loom large as models continue to be applied to more open-ended tasks and higher-stakes domains.

To address these challenges, we propose to use *models themselves* to help convert human preferences into automated decision-making systems. In this paper, we introduce **generative active task**

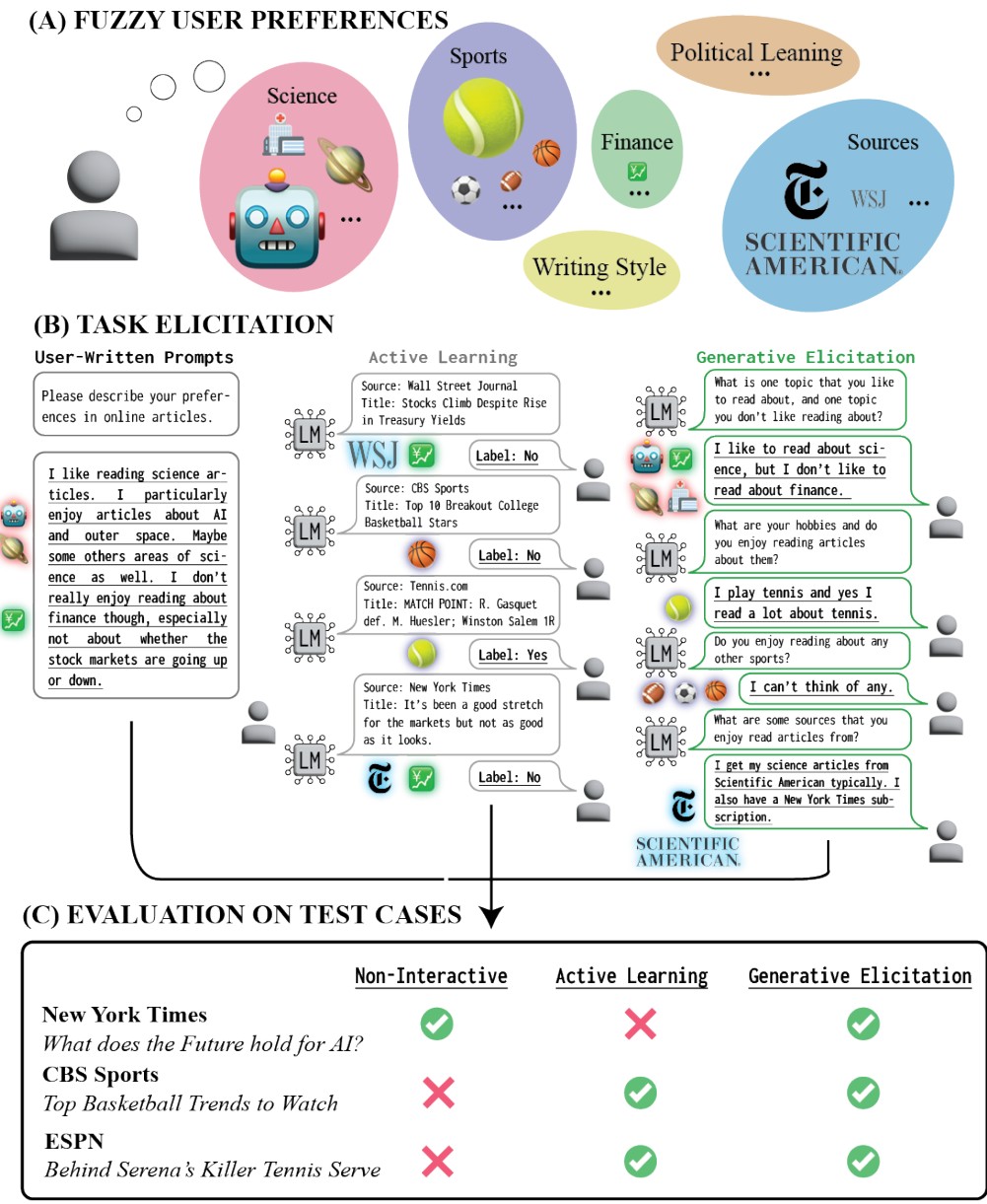

Figure 1: **Generative Active Task Elicitation (GATE) elicits user preferences through interactive, free-form questions, which can then be used in downstream decision-making.** Unlike non-interactive elicitation approaches (e.g., prompting), which rely entirely on the human to elucidate their preferences, generative elicitation is better able to probe nuances of human preferences. Unlike active learning approaches, generative elicitation can ask more generic, free-form questions. The three parts of this figure illustrate: **(A) Fuzzy user preferences:** A user wishes to translate their fuzzy preferences for how a task should be performed into a specification for a machine learning model. This is challenging because users lack perfect introspection, preferences can be difficult to specify in language, the specification needs to anticipate tricky real-world edge cases, and models may misgeneralize from provided examples or instructions. **(B) Task elicitation:** We consider various ways of eliciting these fuzzy preferences from users, including non-interactive prompting, active learning, and generative elicitation (GATE). **(C) Evaluation:** We evaluate methods on a held-out test set, scoring how well a language model predicted the true decisions made by the user.

**elicitation (GATE)**, a learning framework in which models elicit and infer user preferences through open-ended interaction. We describe several techniques for leveraging LMs to perform GATE—for example, by asking informative open-ended questions or generating edge cases for users to label. We evaluate these methods in three domains: email validation, content recommendation, and moral reasoning.[1] In pre-registered experiments, we find that LM-based task elicitation often yields more accurate models than existing prompting or active learning techniques while requiring comparable (or less) mental effort from users and surfacing novel considerations.

In summary, this paper introduces a new learning framework (GATE), a family of methods that perform GATE using pre-trained language models, and experimental evidence showing that these methods outperform existing prompting and labeling methods. Our results show that interactive, language-based task elicitation is a flexible and powerful tool for building personalized models, capable of overcoming many challenges inherent in prompt- and example-based methods.

## 2 Learning as Task Elicitation

### 2.1 The Task Elicitation Framework

We study the problem of efficiently training a machine learning model to perform a task of interest. Throughout this paper, we use **task** to refer generically to any function $f : x \mapsto y$ that maps inputs $x$ to outputs $y$. When building a personalized website recommendation system, for example, $x$ are websites and $y$ are user preference scores for that website. Because different users may prefer different content, each user's individual preferences specify a distinct task: *content recommendation for Pat* and *content recommendation for Avery* are different tasks within the **domain** of content recommendation (Ziegler et al., 2020). To build such a model, we must collect some **task specification** from a human user (e.g., revealing what websites they are interested in). As noted above, current learning approaches admit a wide variety of specification types, including collections of labeled examples, natural language instructions, or combinations of the two. What makes one type of specification preferable to another? Ideally, we would like specifications that are both (1) easy for humans to create and (2) informative to learners, enabling them to model human preferences accurately. Abstractly, we seek a framework for gathering and learning from specifications that optimizes an objective:

$$\alpha \cdot \text{specification cost} + \beta \cdot \text{human–predictor alignment} \tag{1}$$

where **specification cost** measures human time and mental effort, **human–predictor alignment** measures the extent to which model choices agree with choices the human would have made, and $\alpha$ and $\beta$ tradeoff between the two. To formalize this, let $\mathcal{H}_f$ denote a human user whose preferences are represented by a function $f$. We wish to design an **elicitation policy** $\mathcal{E}$ that interacts with $\mathcal{H}_f$ to produce a **task specification** $s$. This specification may then be input to a learning algorithm to produce a model $\hat{f}(s)$. Then, letting $C(\cdot)$ denote a scalar measure of specification cost, and $A(\cdot, \cdot)$ denote a measure of alignment between two predictors, we wish to minimize (in expectation over the population of human users):

$$\mathbb{E}_{\mathcal{H}_f} \mathbb{E}_{s \sim \mathcal{E}(\mathcal{H}_f)} \big[ \alpha \cdot C(s) + \beta \cdot A(f, \hat{f}(s)) \big] . \tag{2}$$

Here, $C$ might measure the number of words the user typed to produce the specification $s$, while $A$ might measure model–predictor agreement at the level of individual predictions from some population: $A(f, \hat{f}) = \mathbb{E}_x \| f(x) - \hat{f}(x) \|$. In general, appropriate definitions of $C$ and $A$ are domain-dependent; in this paper, our experiments compare the alignment of different predictors at a fixed cost. Evaluation of cost, alignment, and tradeoffs between them are discussed more in Section 5.

### 2.2 Existing Learning Paradigms in the Task Elicitation Framework

Several existing frameworks for learning and task specification can be described within the framework given above. Understood as task elicitation procedures, existing frameworks differ along two key axes (visualized in Table 1): their level of *interactivity* and their level of *flexibility*. In interactive

---

[1] While this paper focuses on language-based elicitation procedures, we note that generative active task elicitation is modality-agnostic and could be applied to other settings (e.g., speech-based or multimodal models).

| | Passive | Interactive |
|---|---|---|
| Example-based | Supervised learning | Pool-based active learning |
| Free-form | Prompting | Generative active task elicitation (ours) |

Table 1: Breakdown of different types of task elicitation methods.

elicitation methods, queries can change depending on user responses (e.g., querying for the most useful information based on what is known thus far) while passive elicitation methods expect the user to provide specifications in a single shot. Example-based specification methods ask users to label a set of examples, while free-form elicitation approaches are less restrictive, allowing the user to provide a much wider range of inputs, including natural language instructions and explanations.

**Supervised learning: passive, example-based**   In the most common supervised learning setup, the elicitation policy $\mathcal{E}$ simply instructs the human user $\mathcal{H}_f$ to generate a collection of labeled (input, output) pairs, after which $\hat{f}(s)$ is produced by fitting or fine-tuning a learned model using standard algorithms. This is an *example-based* process because the specification is provided via labeled examples and is *passive*, as the model does not interactively query the user to label additional data.

**Active learning: interactive, example-based**   In active learning, the elicitation policy is interactive. Users first assemble a fixed pool of unlabeled inputs $x$. Next, $\mathcal{E}$, selects from this pool an example whose label would be most informative. The user $\mathcal{H}_f$ provides a label for this example, then $\mathcal{E}$ selects the next-most-informative example, and so on (Cohn et al., 1994; Dagan & Engelson, 1995; Lewis & Gale, 1994; Settles, 2009). Finally, $\hat{f}(s)$ is trained as in supervised methods. Optimal experiment design methods (Emery & Nenarokomov, 1998) may be viewed as generalizations of this paradigm in which inputs $x$ are generated rather than selected. *Interactive* processes enable the model to query for examples that may resolve uncertainty or ambiguity in the task specification (Tamkin et al., 2022b).

**Prompting: passive, free-form**   Modern pre-trained models allow for specifying tasks in more flexible ways than simply labeling examples. For example, models can be conditioned with a *prompt* describing the user's intended task in natural language (Brown et al., 2020b), or even a mix of language and image inputs (Alayrac et al., 2022). As with supervised learning, the labeling policy $\mathcal{E}$ here is simply an instruction to write a natural language task description $(s)$, but the final predictor $\hat{f}(s)$ is produced by passing $s$ to a pre-trained language model.

## 3   GENERATIVE ACTIVE TASK ELICITATION

All of the methods above have important drawbacks: the burden typically falls upon the user to ensure that prompts or example sets are truly comprehensive specifications of the task, as any lack of clarity in the prompt could lead to task ambiguity (Tamkin et al., 2022a), resulting in undesired behavior during deployment. Resolving task ambiguity by crafting better prompts is challenging and time-consuming due to the difficulties of articulating nebulous personal preferences and anticipating edge cases that will emerge during deployment time.

However, one quadrant of Table 1 is not occupied by any of the aforementioned approaches: there is currently no method that leverages both the flexibility of a free-form specification, while using interaction to resolve uncertainty. We explore whether it is possible to combine the flexibility and richness of prompting-based specifications with the advantages of interactive methods such as active learning, by having a model interactively query users for these rich specifications. We term this family of methods **generative active task elicitation (GATE)**.

### 3.1   METHODS FOR GATE

The effectiveness of language models (LMs) for understanding and producing free-form text suggests that they may be capable of eliciting and understanding user preferences. In this paper, we thus experiment with a family of GATE methods in which LMs serve as the backbone for both the elici-

tation policy $\mathcal{E}$ and the predictor $\hat{f}(s)$.[2] See Figure 1 for examples. In particular, we implement the elicitation policy $\mathcal{E}$ by prompting an LM to ask the user questions while conditioning on the history of previous questions and answers. To make predictions $\hat{f}(s)$, an LM is prompted to predict a label conditioned on an input $x$ and a complete elicitation transcript $s$ provided as input. We experiment with several different information gathering policies, realized by simply prompting an LM to ask different kinds of questions:

**Generative active learning** The LM generates example inputs for the user to label. This approach has the advantage of providing concrete scenarios to the user, including some they may not have considered otherwise. For example, for content recommendation, the LM might generate an article such as: *Are you interested in the following article? The Art of Fusion Cuisine: Mixing Cultures and Flavors [...]* .

**Generating yes-or-no questions** We restrict the LM to generating binary yes-or-no questions. This approach enables the model to elicit more abstract preferences while still being easy for the user to answer. For example, the model might probe a user's preferences by asking: *Do you enjoy reading articles about health and wellness?*

**Generating open-ended questions** The LM generates arbitrary questions requiring free-form natural language responses. This enables the LM to elicit the broadest and most abstract pieces of knowledge at the potential cost of being overly broad or challenging for the user to answer. For example, the LM might generate the question: *What hobbies or activities do you enjoy in your free time [...], and why do these hobbies or activities captivate you?*

The user is not constrained in their response in any of the above settings; they are free to provide as much detail as they want. We present example elicitation transcripts for each policy in Figure 8.

## 4 EXPERIMENT SETUP

We consider tasks in three different domains to evaluate our generative active task elicitation methods. A common feature of these domains is that they do not feature a single correct behavior that could be learned during LM pre-training; instead, models must elicit an individual human's preferences in order to make accurate predictions. We allow each human user to interact open-endedly with an elicitation policy $\mathcal{E}$ for five minutes. Next, humans and learned models $\hat{f}(s)$ independently label a set of held-out examples. Finally, we measure agreement between humans and learned predictors. See Figure 8 for examples of environments and dialogues.[3]

### 4.1 DOMAINS AND DATASETS

**Content Recommendation** We consider the domain of online article recommendations, where user preferences vary widely. Models are evaluated on their ability to predict whether a user would like to read a given held-out article. These test cases are taken from popular online newspaper and magazine articles collected by the authors. We provide a website name, article title, and a short description for each test case.

**Moral Reasoning** Moral preferences can be deeply personal and vary significantly across people and cultures. As a test-bed for eliciting moral values, we consider the question of when (if ever) it is ethical to steal a loaf of bread. During evaluation, models are presented with textual descriptions of scenarios and asked to predict whether users will judge it appropriate to steal a loaf of bread. These test cases are constructed manually by the authors.

**Email Verification** Last, we consider the problem of eliciting requirements for a software engineering task. Specification is especially challenging in software engineering due to the many edge

---

[2]However, we emphasize that our method is not specific to language models or natural language and could potentially be applied to other settings such as images, speech, or multimodal models.

[3]Link to the preregistration of experiments and analyses will be made available upon publication.

cases developers need to anticipate and account for. In particular, we focus on specifying requirements for email address validation, where people have varied preferences over how long emails can be, how many subdomains they may possess, and which special characters are allowed, among other factors. Models are evaluated on their agreement with users about the validity of a set of held-out emails; this test set is again manually constructed by the authors.

## 4.2 HUMAN INTERACTION

Human participants in these experiments were recruited from English-speaking users of Prolific. For the email validation task, we additionally recruited participants from several computer science programs at US universities. We recruited 20–30 participants for each domain-method pair (6 elicitation methods across 3 domains), for a total of 388 participants. Participants were paid an average of \$12/hr. Our experiments received IRB approval. The breakdown of the number of participants allocated to each scenario and method can be found in Appendix B.1. Details of the user interface used in experiments may be found in Appendix B.2.

## 4.3 MODELING DETAILS

We use the GPT-4 model (`gpt-4-0613` snapshot) (OpenAI, 2023) to both elicit user preferences (the elicitation policy $\mathcal{E}$) and make predictions based on the elicited preferences (the predictor $\hat{f}(s)$). To elicit user preferences, we prompt GPT-4 with a domain description and the current interaction history, and ask it to generate an informative but easy-to-answer edge case (for generative active learning) or question (for generative yes-or-no questions and generative open-ended questions). To make predictions, we prompt GPT-4 with the task specification $s$ and a test sample $x$ and ask it to generate a prediction for the test sample. The full text of the prompts can be found in Appendix A.

## 4.4 BASELINE METHODS

We compare GATE with several baseline approaches for specifying tasks. Here, the elicitation policy $\mathcal{E}$ is not parameterized by an LM, but constructed by the user and/or a pool of examples.

**Supervised learning**   We consider supervised learning as a baseline, as described in Section 2.2. We randomly present participants with questions from a large pool of examples and ask them to annotate up to the time limit. We study this approach exclusively in the content recommendation domain because pools of examples are not readily available in the other two domains. We use the Microsoft News Dataset (Wu et al., 2020) as our pool for this domain, a dataset of 160k news articles with descriptions.

**Pool-based active learning**   As a baseline active learning approach, we consider a pool-based active learning approach, as described in Section 2.2. For the elicitation policy, we use the diversity-based sampling approach of Margatina et al. (2023); we first cluster the examples using a Sentence-BERT embedding model (Reimers & Gurevych, 2019) into 15 different clusters, then iteratively ask questions from each cluster in a round-robin fashion, up until the time limit.[4] This baseline is intended to capture the difficulty of selecting informative examples from a pool of unlabeled examples relative to generating informative examples from scratch. As with supervised learning, we study this approach exclusively in the content recommendation domain.

**User-written prompts**   As a baseline that does not use interactive elicitation, we ask participants to write a short paragraph describing their preferences for the task. We then use the text of this paragraph to prompt a model to make decisions. This baseline is intended to capture the difficulty of specifying preferences in writing, both in terms of the effort it takes to write the paragraph and the difficulty of writing a paragraph that fully specifies one's preferences.

---

[4]Margatina et al. (2023) explored several different popular active learning sampling approaches for in-context learning (including random, uncertainty, and diversity sampling) and found little difference in empirical performance between them. We also ran exploratory model-model experiments in our domains and found no significant difference between these three sampling strategies. See details in Appendix D.

### 4.5 EVALUATION AND METRICS

We measure how well models can predict the probability that users will answer questions a certain way. Specifically, we prompt the model to output a real-valued *probability* of answering *yes* to the question, as opposed to a binary yes/no decision. To do so, we prompt the model with the interaction history $s$ as a single test case, then ask the model to predict the probability that a user would answer "yes" to the test case. This probability is outputted in token space as a number between 0.0 and 1.0, similar to past work (Branwen, 2020; Lin et al., 2022).[5] We also discuss and report a classification-based metric in Appendix C.1. Given these predicted probabilities, we compute:

**Area under the $p(\textbf{correct})$-time curve**  We define $p(\text{correct})$ as the probability the model assigns to the user-preferred answer (see Section 4.5). For example, if the model outputted `0.8` for a given question, then $p(\text{correct})$ would be $0.8$ if the user's answer were "yes" to the same question, and $0.2$ if the user's answer was "no". We select this metric instead of accuracy because guessing the user's preferences may not always be possible, and modeling this uncertainty is useful.

However, we do not just care about the total information elicited, but about *how quickly* good information is elicited. To do this, we compute the average change in $p(\text{correct})$ after *every minute* of human elicitation time (conditioning on the state of the transcript at that time). This produces a curve where the $x$-axis is time, and the $y$-axis is the average change in $p(\text{correct})$. The area beneath this curve is a second metric we consider. Note that the final data point of each $p(\text{correct})$ curve may not reach 5 minutes because we subtract out the latency from the language modeling API; to account for this, we extend the final accuracy to the 5-minute mark before computing the area.

**Rating of perceived effort across elicitation policies**  In addition to these performance-based metrics, we also ask users to rate how difficult they found the elicitation process to be.

Specifically, we asked users "*How mentally demanding was writing your answer?*" in the non-interactive-elicitation setting, and "*How mentally demanding was interacting with the chatbot?*" in all elicitation settings (which include all other settings from Section 2.2). The "mentally demanding" wording was taken from the NASA TLX (Hart & Staveland, 1988). The question was assessed via a Likert scale from 1 (Very Little) to 7 (Very High). We also consider several additional questions to assess other usability tradeoffs. See Appendix E for the full list.

## 5 RESULTS

Evaluation results are shown in Figures 2 and 3. Additional results can be found in Appendix C. These results show that GATE methods...

**...are successfully able to elicit human preferences.** Overall, GATE improves over no elicitation, where the model is prompted to make decisions before any user interaction. This is the case across all domains studied (a positive score in Figure 2), with significance at the 0.05 level for all but the email domain, where only generative active learning was significant.

**...are comparable to or better than other elicitation methods.** In the majority of settings (6/10 for absolute, 7/10 for AUC), GATE elicitation methods improve over user-written prompts. In particular, generative yes/no questions improve over user-written prompts in every setting studied (although we lack enough power to assess significance). Furthermore, in the content recommendation setting, GATE elicitation methods (particularly generative open-ended questions) significantly improve over supervised learning and pool-based active learning.

**...are equally or less mentally demanding than user-written prompts.** As shown in Figure 3 (left), users generally find interactive elicitation methods to be less mentally demanding, especially ones that involve labeling samples or answering yes/no questions, than non-interactive prompting.

---

[5]While there may be other ways one might make predictions with these models, we found them lacking for a variety of reasons. First, we conducted pilot experiments prompting the LM to predict binary yes/no decisions; however, we found this resulted in skewed predictions where the LM would predict one of 'yes' or 'no' for the entire test set, perhaps due to miscalibration of the model's implicit decision threshold. Second, we found that LMs are generally less reliable when generating confidence values in log space. Finally, we cannot directly take the token probabilities from GPT-4 using the OpenAI API.

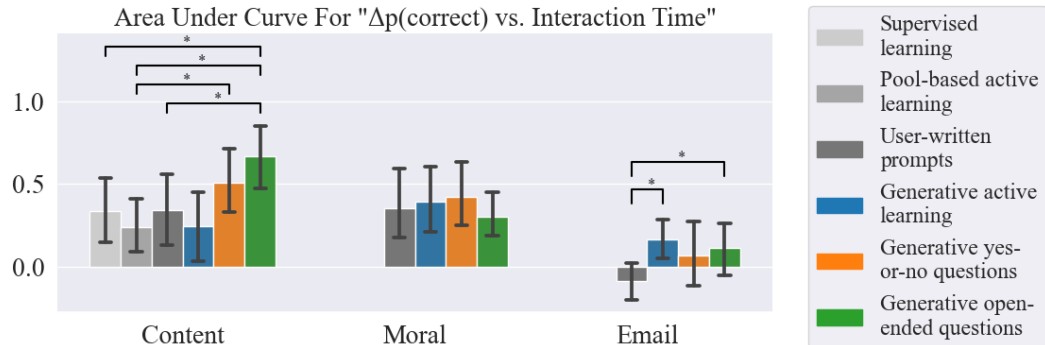

Figure 2: **Across three domains, our LM-prompting implementations of GATE are generally able to elicit human preferences beyond baseline supervised learning, active learning, or human-written prompts.** We measure the Area Under the "$\Delta p(correct)$ vs. Interaction time" Curve, which gives us a time-normalized metric for how well (and how quickly) each elicitation method is at aligning with human preferences. While GATE methods generally outperform the baseline methods as well as no interaction (represented by a $\Delta p(correct)$ of 0), we are only able to establish statistical significance between GATE and baselines in the content recommendation and email verification domains.

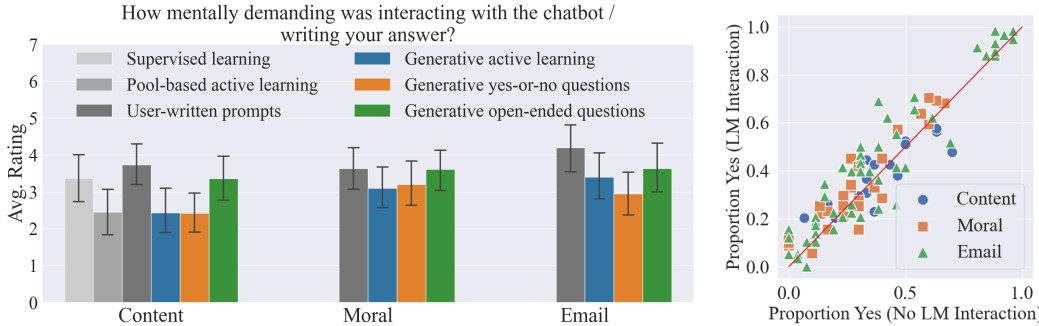

Figure 3: **Left: GATE methods are equally or less mentally demanding than other methods.** We plot the perceived mental demand across methods and domains (higher = greater mental demand).
**Right: Language model elicitation does not shift human preferences.** We plot the proportion of participants who answered "yes" to each test question, comparing no LM interaction (user-written prompts) to LM interaction (GATE) elicitation. The red line is the $y = x$ curve, which serves as a guideline to see how well humans' no-LM interaction preferences align with their preferences post-LM interaction (if they align perfectly, the points should fall along this curve). We see that the points generally hover around this curve.

**Does language model elicitation influence user preferences?** Human preferences may shift when interacting with language models for a variety of reasons. For example, past work has studied *auto-induced distributional shift*, where machine learning models shift human behavior to be easier to predict (Krueger et al., 2020). To investigate whether this occurs in our experiments (or indeed if different elicitation methods induce different human preferences for any other reason), we compare the distribution of human labels on test samples from the three GATE methods with those from the user-written prompt experiments to see whether interacting with language models influences users' subsequent judgments. As seen in Figure 3 (right), we see no such effect.

# 6 OTHER RELATED WORK

## 6.1 ELICITING DESCRIPTIONS OF PREFERENCES

A fundamental challenge across many fields is how to obtain information about people's nebulous thoughts, preferences, and goals. In psychology and cognitive science, *protocol analysis* describes methods for how to obtaining and analyze verbal reports from subjects about cognitive processes including via *think-aloud protocols* (Ericsson & Simon, 1980; Ericsson, 2017). In software usability analysis, similar techniques are used to assess the usability and limitations of existing software (Henderson et al., 1995), and for broader applications in the areas of survey, questionnaire, and focus group design (Malhotra, 2006; Lietz, 2010; Krosnick, 2018; Krueger & Casey, 2002). High-quality

verbal reports can be challenging to obtain, however, and *requirements elicitation* studies methods for gathering information even when it is challenging for users to fully understand or anticipate their own needs or describe their preferences in clear, unambiguous language (Christel & Kang, 1992; Goguen & Linde, 1993; Coughlan & Macredie, 2002; Zowghi & Coulin, 2005; Pacheco et al., 2018). In our work, we explore whether language models could take the place of human researchers in surfacing these insights from people or even other language models.

## 6.2 COMPUTATIONAL MODELING AND QUERYING OF PREFERENCES

Many works attempt to computationally describe or query human preferences. Preference modeling techniques study people's *revealed preferences* (Samuelson, 1948), as well as their *stated preferences* (Kroes & Sheldon, 1988), including preferences refined through deliberation (Gutmann & Thompson, 2004). Methods for eliciting preferences span a wide variety of research areas including conjoint analysis (Green & Srinivasan, 1978), multiple-criteria decision making (Greco et al., 2016), multi-armed bandits (Robbins, 1952) and dueling bandits (Yue et al., 2012), Bayesian methods (Chajewska et al., 2000), recommender systems (Aggarwal et al., 2016; McAuley, 2022), robust optimization (Vayanos et al., 2020), optimal experiment design (Emery & Nenarokomov, 1998), (cooperative) inverse reinforcement learning (Ng et al., 2000; Hadfield-Menell et al., 2016), question generation (Mulla & Gharpure, 2023), and generative modeling (Zhu & Bento, 2017).

Perhaps most relevant to our work is active learning, a major subfield of machine learning that centers on how models can choose useful data points to learn from. Active learning has traditionally focused on pool-based methods, which choose points to label from a fixed reservoir (Lewis & Catlett, 1994; Settles & Craven, 2008; Settles, 2009; Houlsby et al., 2011). Recently, Tamkin et al. (2022b) found that the well-calibrated uncertainty scores of pretrained models can be used during active learning to clarify the user's task preferences—for instance, by choosing examples that distinguish which of two correlated features are important for the task. We extend this line of investigation to the generative setting, clarifying user intent by querying a user with *generated* examples and questions.

## 6.3 TASK AMBIGUITY AND UNDERSPECIFICATION

A growing body of work explores how tasks in machine learning can be underspecified or ambiguous. In particular, *task ambiguity* (Finn et al., 2018; Tamkin et al., 2022b) arises when more than one task is consistent with the inputs to the model (e.g. the natural language prompt or provided examples). One stream of work here investigates spurious correlations (Geirhos et al., 2020), a form of task ambiguity where the network learns unwanted associations between features in the input data and the task label (Nagarajan et al., 2021; Sagawa et al., 2019; Srivastava et al., 2020; Sagawa et al., 2020). Such underspecified training pipelines can lead to unpredictable and undesired behavior during deployment and potentially dangerous real-world consequences (D'Amour et al., 2022). As recent models can accept richer specifications, such as natural language prompts, task ambiguity can arise from other sources, such as incomplete or suboptimal natural language descriptions of the task (Tamkin et al., 2022b). In this work, we find that language models can often resolve their own task ambiguity in these instances by asking informative questions of the user.

## 7 DISCUSSION AND CONCLUSION

We introduced the GATE framework to interactively elicit preferences from human users with free-form queries and answers. We presented initial evidence that language models can successfully implement GATE to elicit human preferences (sometimes) more accurately and with less effort than supervised learning, active learning, or prompting-based approaches.

There are many ways to expand on our implementation of GATE: Future work may explore more principled methods for elicitation besides simple prompting; for example, explicit notions of uncertainty or disagreement sampling could be used in conjunction with the free-form generation enabled by generative language models, taking inspiration from the active learning literature. Second, larger models may be more capable elicitors: future work can explore scaling laws for elicitation. Finally, many real-world tasks are more complex than those we study here; applications such as software design and legal and medical decision-making present a richer set of constraints and edge cases. These applications thus offer a rich space of possible extensions of GATE.

ETHICAL CONSIDERATIONS

Our work presents several potential ethical benefits and risks.

There are many potential benefits of machines that can better elicit and understand human preferences. For example, by making it easier for software designers to incorporate nuanced user preferences, GATE may empower people with rare preferences or preferences that have historically not been considered when building software systems. In addition, improving the effort-performance ratio, especially by requiring less user typing, may help make language models more accessible to users with less time, familiarity with language models, or physical ability to use such systems.

However, this direction carries risks as well. In particular, work on *thin slicing* (Ambady & Rosenthal, 1992) has demonstrated that small amounts of information about a user can sometimes be used to predict a broader range of personal characteristics, raising potential privacy considerations. The interactive nature of GATE also risks increasing *automation bias* (Goddard et al., 2012), where users place undue weight on a model's predictions. However, further work is necessary to establish if or when these risks are more significant for GATE than for prompting-based approaches to steering language models.

REPRODUCIBILITY

We will open-source all code used in creating GATE methods, constructing the user interface, and conducting the results and analysis. We will also release the pre-registration for our experiments. All prompts we used for querying GPT-4 in the decision-making and elicitation phases, and all instructions we presented to the user, can be found in the Appendix. In all cases, we queried GPT-4 with temperature 0 for replicability of experiments.

We also note that the model we use is a closed-source model whose versions are periodically deprecated. This may hinder reproducibility, and we may explore open-source models in the future.

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

# A    GATE PROMPTS

## A.1    ELICITATION PROMPTS

Below, we list the prompts we use for generating elicitation queries (to instantiate the elicitation policy $\mathcal{E}$) for the various GATE methods.

**Generative active learning**    The prompt for generative active learning can be found below. Bracketed, italicized text are placeholders, e.g. for the elicitation transcript (so far). Color-coded text is domain-specific, e.g. the blue text represents text for the content recommendation domain, orange text represents text for the moral reasoning domain, and the green text represents text for the email validation domain.

---

Your task is to *[ learn what topics a user is interested in reading online article about. People's interests are broad, so you should seek to understand their interests across many topics; in other words, go for breadth rather than depth. Do not assume a user has given a complete answer to any question, so make sure to keep probing different types of interests | learn under what conditions the user would believe it is or is not ethical to steal a loaf of bread | learn what rules a user believes a valid email address format must adhere to (e.g. for developing a regex format checker) ]*.

Come up with a potential edge case to learn as much information as you can about what their desired behavior should be under different circumstances.
Make sure the edge case addresses different aspects of the system than the edge cases that have already been considered.

An example edge case is: *[ Are you interested in the following article? Website Name: TechTalkDaily
Title: 10 Must-Have Gadgets for 2023: A Comprehensive Guide
Description: Discover the top 10 gadgets that will revolutionize your daily life in 2023. From smart home devices to the latest wearables, we've got you covered with our comprehensive guide. | Situation: Is it ethical to steal a loaf of bread if you are starving to death and the bread would otherwise go to waste? | Should the following email be accepted? username@example.com ]*

Current cases:
*[ Elicitation transcript ]*

Generate the most informative edge case that, when answered, will reveal the most about the desired behavior beyond what has already been queried for above. Generate the edge case in the following format, and nothing else: "*[ Are you interested in the following article? [edge case] | Situation: [edge case] | Should the following be accepted? [edge case] ]*"

---

**Generating Questions**    The prompt for generating both yes-or-no and open-ended questions can be found below. Once again, bracketed, italicized text are placeholders and color-coding indicates text for specific domains.

---

Your task is to *[ learn what topics a user is interested in reading online article about. People's interests are broad, so you should seek to understand their interests across many topics; in other words, go for breadth rather than depth. Do not assume a user has given a complete answer to any question, so make sure to keep probing different types of interests | learn under what conditions the user would believe it is or is not ethical to steal a loaf of bread | learn what rules a user believes a valid email address format must adhere to (e.g. for developing a regex format checker) ]*.

Previous questions:
*[ Elicitation transcript ]*

Generate the most informative *[ yes/no question | open-ended question ]* that, when answered, will reveal the most about the desired behavior beyond what has already been queried for above. Make sure your question addresses different aspects of the implementation than the questions that have already been asked. At the same time however, the question should be bite-sized, and not ask for too much at once. Phrase your question in a way that is understandable to non-expert humans; do not use any jargon without explanation. Generate the *[ yes/no question | open-ended question ]* and nothing else:

---

## A.2 Decision-making Prompts

Given an elicitation transcript, we prompt the LM as follows to serve as a predictor $\hat{f}(s)$ to make decisions about each test case.

**Content Recommendation**   The prompt for content recommendation can be found below. Bracketed, italicized text are placeholders, for the elicitation transcript and the test case sample.

---
A user has a particular set of preferences over what articles they would like to read. They have specified their preferences below:
*[ Eliciation transcript ]*

Based on these preferences, would the user be interested in reading the following article? Answer with a probability between 0 and 1, where 0 means "definitely not interested" and 1 means "definitely interested". Only output the probability and nothing else. If uncertain, make your best guess.
*[ Test case sample ]*

---

**Moral Reasoning**   The prompt for moral reasoning can be found below. Once again, bracketed, italicized text are placeholders.

---
A user has a particular ethical code they follow. The following response(s) represent when this user would believe it is ethical to steal a loaf of bread.
*[ Eliciation transcript ]*

"Based on these response(s), would the user believe it is ethical to steal a loaf of bread in the following situation? Answer with a probability between 0 and 1, where 0 means "definitely not" and 1 means "definitely yes". Only output the probability and nothing else. If uncertain, make your best guess.
*[ Test case sample ]*

---

**Email Validation**   The prompt for email validation can be found below. Once again, bracketed, italicized text are placeholders.

---
A user has a particular format of emails that they believe to be valid. The following answer(s) represent this user's preferences of whether these emails adhere to their desired format.
*[ Eliciation transcript ]*

Based on the user's preferences, does the following email adhere to the user's desired format? Answer with a probability between 0 and 1, where 0 means "definitely not" and 1 means "definitely yes". Only output the probability and nothing else. If uncertain, make your best guess.
*[ Test case sample ]*

---

# B   Experimental Details

## B.1   Number of Participants

The number of participants we recruited for our study, for each elicitation method and domain, can be found in the table below.

|  | Content Recommendation | Moral Reasoning | Email Validation | Total |
|---|---|---|---|---|
| Supervised learning | 30 | - | - | 30 |
| Pool-based active learning | 31 | - | - | 31 |
| Prompting | 30 | 30 | 26 | 86 |
| Generative active learning | 30 | 30 | 20 | 80 |
| Generative yes-or-no questions | 31 | 30 | 19 | 80 |
| Generative open-ended questions | 31 | 31 | 19 | 81 |
| Total | 183 | 121 | 84 | 388 |

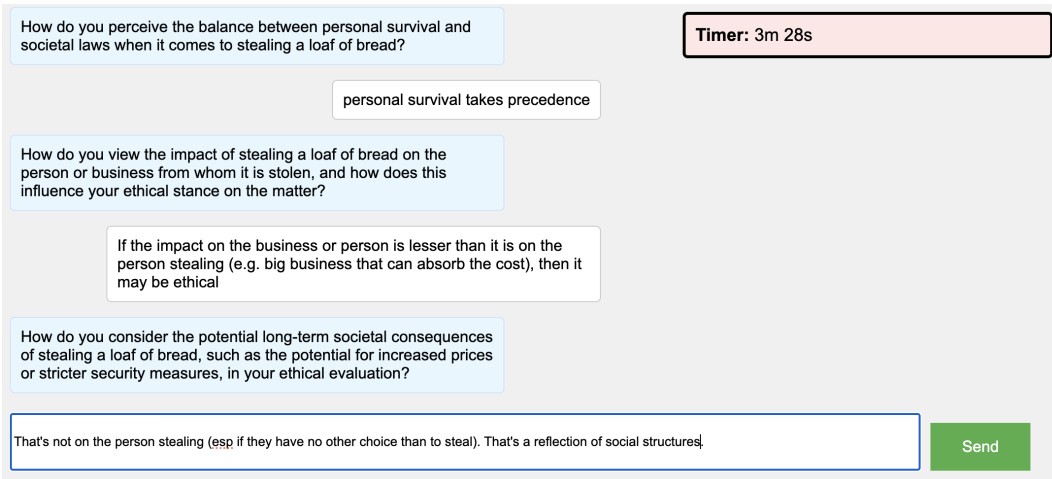

Figure 4: Chatbot UI built for elicitation phases of GATE methods, supervised learning, and pool-based active learning.

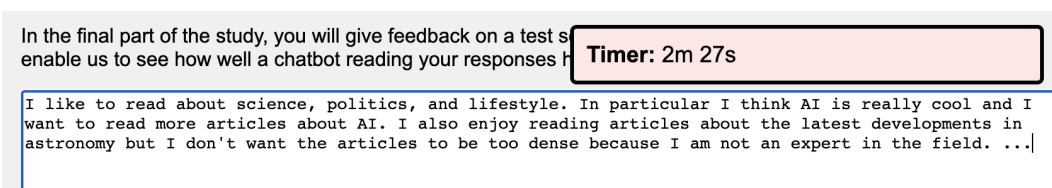

Figure 5: Text-input UI built for elicitation phase for prompting.

## B.2 USER INTERFACE DETAILS

Details about the UI we built for our experiments can be found below. Recall that the human studies proceeded in two parts: elicitation, followed by decision-making.

### B.2.1 ELICITATION

For supervised learning, pool-based active learning, and the GATE methods, we had participants respond to a series of queries using the chatbot interface (Figure 4). For prompting, we had participants input a task description using the text-input interface (Figure 5).

The instructions for this phase can be found below.

**Supervised Learning / Pool-based Active Learning** We present users with the following instructions for both supervised learning and pool-based active learning. Bracketed, italicized text represent placeholders for domain-specific text. *[ Domain instructions ]* is a placeholder for the top-level instructions for each domain (see Table 2). Otherwise, blue text represents text for the content recommendation domain, orange text represents text for

| Content | We are testing a system for understanding people's interest in reading different kinds of online articles. |
| --- | --- |
| | For example, you might be interested in articles about some topics, but not about others. |
| Moral | We are testing a system for understanding people's fuzzy intuitions and preferences. |
| | In this experiment, we'll be capturing your moral intuitions about the act of stealing a loaf of bread, and whether there are certain cases where stealing may be morally permissible. |
| Email | We are testing a system for understanding people's fuzzy intuitions and preferences. |
| | In this activity, we're going to be looking at different strings of text and you'll be deciding if they look like they could be an email address or not. For example, most people would agree that "username@domain.com" looks like an email address, while "n12z5lFEN4" does not. However, the rules for what can be an email address can be very unusual, so what we're really interested in is your intuition on what an email address could look like. |
| | **Important:** We are not asking you to determine the rules for a \*good\* email address, or a \*real (non-spam)\* email address. We are simply asking about your intuition as to why certain strings look like email addresses and certain strings do not. |
| | **Tip:** in an email such as username@cs.stanford.edu, "username" is called the local-part of the email, while "cs.stanford.edu" is the domain. Furthermore, "cs" is a subdomain, and "edu" is a top-level domain. |

Table 2: Domain-specific instructions presented to users for the elicitation phases.

the moral reasoning domain, and green text represents text for the email validation domain.

> [ Domain instructions ]
>
> Try to answer in a way that accurately and comprehensively conveys your preferences, such that someone reading your responses can understand and make judgments as close to your own as possible. Feel free to respond naturally (you can use commas, short phrases, etc), and press [enter] to send your response. Note that the chatbot technology is imperfect, and you are free to avoid answering any questions that are overly broad or uncomfortable. When interacting with the chatbot, please avoid asking follow-up questions or engaging in open-ended dialogue as the chatbot is unable to respond to you.
>
> **Note:** The chatbot will stop asking questions after 5 minutes, after which you can send your last response and you will be taken to the final part of the study.
>
> In the final part of the study, you will give feedback on a test set of [ *article headline and descriptions* | *moral situations* | *email addresses* ], which will enable us to see how well a chatbot reading your responses has learned [ *what you like and dislike* | *your moral preferences* | *your email preferences* ].

**Prompting**   We present users with the following instructions for prompting. Similar to above, bracketed, italicized text represent places where we insert domain-specific text.

*[ Domain instructions ]*

To the best of your ability, please explain all details about *[ your preferences of what kinds of online articles you would like to read | your belief of when it is moral to steal a loaf of bread | your intuition of what makes email addresses look like email addresses ]*, such that someone reading your responses can understand and make judgments as close to your own as possible. Try to be as detailed as possible. For example, if you were writing a regex that accepts only email-address-like strings, what might that regex look like? What are permissible / non-permissible symbols and characters, and in what positions?

**Note:** You will have up to 5 minutes to articulate your preferences. Please try to submit your response within that time. After you submit, you will be taken to the final part of the study.

In the final part of the study, you will give feedback on a test set of *[ article headline and descriptions | moral situations | email addresses ]*, which will enable us to see how well a chatbot reading your responses has learned *[ what you like and dislike | your moral preferences | your email preferences ]*.

**GATE methods**   We present users with the following instructions for the three GATE methods (generative active learning, generative yes-or-no questions, generative open-ended questions). Once again, bracketed italicized text represent domain-specific text.

*[ Domain instructions ]*

This chatbot will ask you a series of questions about *[ your preferences of what kinds of online articles you would like to read | your belief of when it is moral to steal a loaf of bread | your intuition of what makes email addresses look like email addresses ]*. Try to answer in a way that accurately and comprehensively conveys your preferences, such that someone reading your responses can understand and make judgments as close to your own as possible. Feel free to respond naturally (you can use commas, short phrases, etc), and press [enter] to send your response. Note that the chatbot technology is imperfect, and you are free to avoid answering any questions that are overly broad or uncomfortable. When interacting with the chatbot, please avoid asking follow-up questions or engaging in open-ended dialogue as the chatbot is unable to respond to you.

**Note:** The chatbot will stop asking questions after 5 minutes, after which you can send your last response and you will be taken to the final part of the study.

In the final part of the study, you will give feedback on a test set of *[ article headline and descriptions | moral situations | email addresses ]*, which will enable us to see how well a chatbot reading your responses has learned *[ what you like and dislike | your moral preferences | your email preferences ]*.

### B.2.2   DECISION-MAKING

For the decision-making phase, we presented users with the following instruction:

**Content Recommendation**

Please indicate whether you would like to read the following articles: "yes" if you would, "no" if you would not.

**Moral Reasoning**

Please indicate whether you think the following situations are morally permissible or not: "yes" if they are, "no" if they aren't.

**Email Validation**

Please indicate whether you think the following strings look like reasonably well-formatted email addresses or not: "yes" if they do, "no" if they don't.

Users are then presented with a list of test samples, and can use radio buttons to select whether each test-case sample is acceptable. See Figure 6.

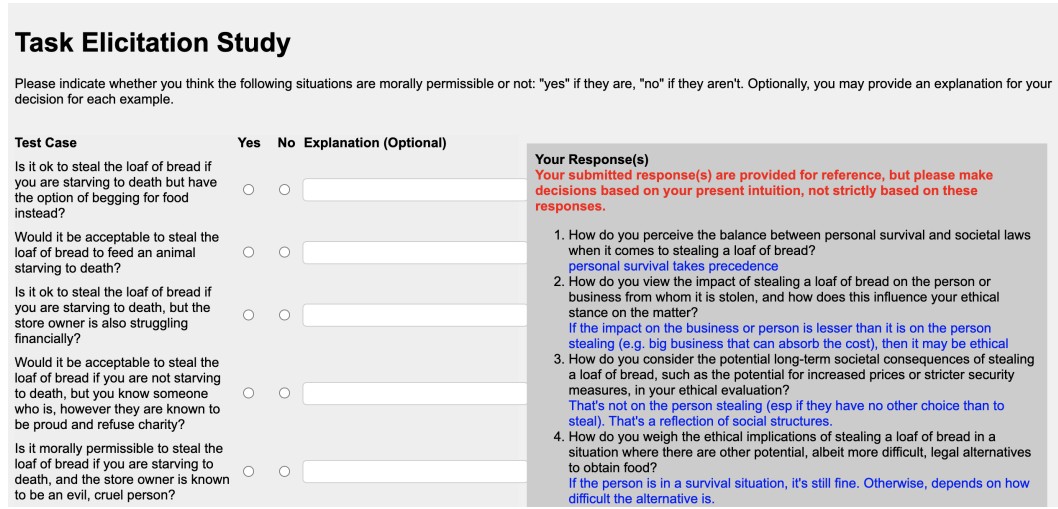

Figure 6: UI for the decision-making phase.

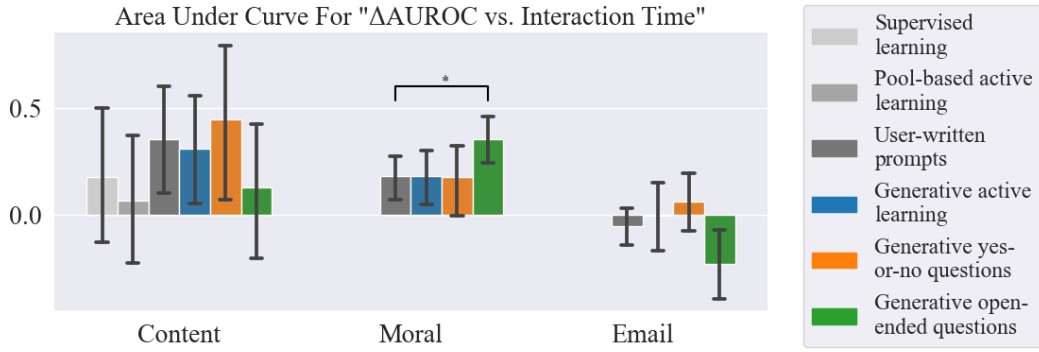

Figure 7: We plot the **Area Under the "ΔAUROC vs. Interaction time" Curve**, which gives us a metric for how well (and how quickly) each elicitation method is at aligning with human preferences. This plot is analogous to Figure 2, only we are using AUROC instead of $p(correct)$ for the alignment metric, which means that we are not measuring uncertainty. We see the same trends hold of GATE methods (generally) beating supervised learning, pool-based learning, and prompting approaches, while also beating no interaction ($\Delta$AUROC $= 0$) using this metric. However, we see generally smaller $\Delta$s over non-interaction using this metric, and higher variances, which make it harder to establish statistical significance.

## C ADDITIONAL RESULTS

### C.1 AUROC RESULTS

We measure AUROC over model-generated probabilities in addition to $\Delta p(\text{correct})$. Figure 7 is the analogous plot to Figure 2, but we measure the improvement in **AUROC** instead of $p(\text{correct})$, over interaction time, rewarding methods that achieve higher improvements in AUROC sooner.

The general trends hold from Section 5: language models can elicit human preferences (beyond no interaction), and language model elicitation is comparable or better than other elicitation baselines. However, unlike the $p(\text{correct})$ metric, the AUROC metric is a simple classification-based metric. Due to potential miscalibration in LMs, making it difficult for them to output well-calibrated probabilities with the same threshold across questions, the overall improvements in this metric are lower (particularly for generative open-ended questions) and the variances are much higher. Thus, we see that it is harder to establish statistical significance using this metric.

## C.2 SAMPLE TRANSCRIPTS

Sample transcripts of users interacting with the various generative active task elicitation methods can be found in Figure 8.

## C.3 ANALYSIS

Here, we present some additional analyses to better characterize the experiments.

**How much variation there is in people's preferences?** Elicitation is only helpful if there is variation in people's preferences; otherwise, a model could simply attain maximum performance by relying on its prior and ignoring the elicited information. To quantify how much variation there is in people's preferences, we compute the entropy in $p(\text{yes})$ for each question across participants. We find that many questions have high entropy while many others have little entropy, for an average entropy of 0.77 bits. Broadly, the results validate that our settings have significant variation in human preferences, enabling models to personalize themselves based on human preferences.

**What kinds of questions did the language models ask?** We show a few examples of the language model questions in Figure 8. As the figure shows, these questions are complex and subtle, often building on the previous questions, representing a broad-based knowledge of the domain as well as possible nuances therein.

**Why does prompting make things worse in the emails domain?** In the emails domain in Figure 2, we observe that user-written preferences slightly decrease performance relative to a no-elicitation baseline. While it is possible this is an effect of noise, we also observe that some participants articulated preferences that were actually different from those they experienced when viewing email addresses. For example, one user wrote "*an email address should finish with .com or co.uk*" yet later decided that "*user@domain.edu*" was an acceptable email address. This indicates that users may not have a clear and comprehensive understanding of their own preferences, especially in more technical domains.

**Can we automate evaluation?** To probe whether evaluation could be automated, we conducted experiments where we *simulated* different human preferences using language models prompted with a diverse set of (automatically-generated) *personas*. These personas varied by domain, but generally contained information about a hypothetical person's preferences within that the domain. For example, in the content recommendation domain, we generated brief biographical sketches of hypothetical people, including their hobbies, interests, and careers, and conditioned GPT-4 on these biographical sketches to generate answers to queries. We found that model could simulate humans well in the content recommendation and email validation domains, but not in the moral reasoning domain. This suggests that while such personas may be a useful guide in some cases, they are not yet sophisticated enough to stand in for real human participants. See Appendix D for more details.

## D MODEL-MODEL EXPERIMENTS

## D.1 METHODS

We explore whether LMs can stand-in for human participants, enabling faster iteration loops and more research in this area. We generate various personas (specified in natural language) for each domain, and prompt LMs to respond to elicitation queries as their persona would.

For each domain, we construct a set of personas as follows:

**Content Recommendation** The personas are constructed by providing a brief biographical sketch of a hypothetical person, and were also constructed by the authors. A sample persona prompt is "*Education: Medical Doctorate. Occupation: Junior Surgeon at a regional hospital. Hobbies: Running marathons, traveling, and learning new languages.*"

**Content Recommendation**

System: Are you interested in the following article?
Website Name: HealthMindBody
Title: The Intersection of Yoga and Mental Health: A Deep Dive
Description: Explore the profound effects of yoga on mental well-being. This comprehensive analysis brings together research, personal stories, and tips for integrating yoga into your mental health routine.
**User: No**
System: Are you interested in the following article?
Website Name: ClimateFutures
Title: The Interplay of Modern

System: Do you enjoy reading articles about advancements in technology and science?
**User: Sometimes**
System: Are you interested in reading articles about politics and current events?
**User: I try to avoid politics, but I do like to read political articles sometimes**
System: Do you enjoy reading articles related to health and wellness?
**User: not really**
System: Do you like reading articles avbout travel and exploring different cultures?

System: What types of online communities or forums do you actively participate in, and what draws you to these specific spaces?
**User: Well, I currently use Reddit for the extreme fine-tuning of my preferences. If a feed is too generic, I won't like it. I go to a lot of specific subreddits.**
System: Can you share some of the specific subreddits you frequent and what topics or discussions within those subreddits you find most engaging?
**User: Let me try using the first five that come to mind:**

**Moral Reasoning**

System: Situation: Is it ethical to steal a loaf of bread if it is from a large corporation that wouldn't notice a small loss, but you are not in immediate physical need, but rather stealing it to feed your family due to being unable to find work?
**User: If I am stealing it to feed my family then I probably wouldn't have a problem with it. If I was stealing it to get back at the corporation that makes it then it would be a problem.**
System: Situation: Is it ethical to steal a loaf of bread if you're in

System: Do you believe it is ethical to steal a loaf of bread if it is to feed a starving child?
**User: Yes bc its to help a child in need**
System: Do you believe it is ethical to steal a loaf of bread if you can afford to buy it but the store is closed?
**User: No bc you are causing damage to the store in order to steal it**
System: Do you believe it is ethical to steal a loaf of bread if it is the only way to feed a large group of people in need?
**User: Yes**

System: How do you perceive the impact of personal circumstances, such as extreme hunger or poverty, on the ethical considerations of stealing a loaf of bread?
**User: I think it's a terrible tragedy when people don't have enough money to eat or feed their families. While stealing is wrong, I don't think stealing bread for survival is immoral, it's survival.**
System: How do you view the role of societal systems and structures in creating situations where someone might feel compelled to steal a loaf of bread, and does this

**Email Verification**

System: Should the following be accepted?
user.name+alias@example.co.uk
**User: no**
System: Should the following be accepted? user_name-123@example.org
**User: yes**
System: Should the following be accepted?
user@subdomain.example.com
**User: sure, but the "example.com" is a bit weird**
System: Should the following be accepted? user.name@123-example.com
**User: sure**
System: Should the following be accepted?

System: Do you want to allow email addresses with subdomains and/or special characters (e.g., "+" or "_") in the username part?
**User: Yes, I want to allow email addresses with subdomains, but not with '+' sign but '_' is okay**
System: Do you want to enforce a specific length limit for the username and domain parts of the email address, or allow any length as long as it is a valid email format?
**User: Yeah, we need a 128 characters limit for username part and same for domain parts**

System: Can you describe what kind of special characters, if any, you believe should be allowed in an email address and where they should be placed?
**User: A dot separating name**
System: What are your thoughts on the use of numbers and capital letters in an email address, and do you believe there should be any restrictions on where they can be placed?
**User: No use of capital letters, email should be insensitive to that**
System: What are your thoughts on the length of an email address? Do



**Generative
Active Learning**  **Generative
Yes / No Questions**  **Generative
Open-Ended Questions**



Figure 8: **Excerpts of real transcripts across the different domains and elicitation methods we investigate.** The System messages are generated by the language model, while the User messages are produced by human participants. Overall, the model is able to generate diverse and contextually-appropriate questions in each setting. See Sections 3.1 and 4.1 for more details on the domains and methods respectively.

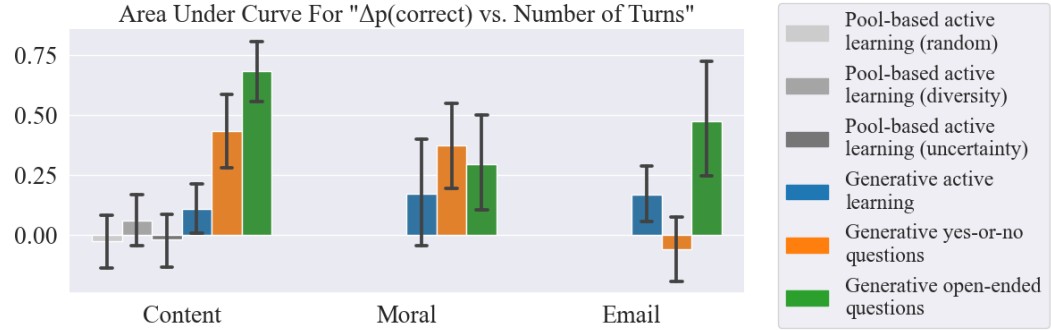

Figure 9: We plot the **Area Under the "$\Delta p(correct)$ vs. Number of Turns" Curve** for model-model experiments. This plot is analogous to Figure 2, only we are using LMs to simulate human users, and we are using number of turns as a proxy for interaction time. We see the same general trends as in Figure 2: GATE methods beat both no elicitation and pool-based active learning.

**Moral Reasoning** We construct a variety of personas with a diverse array of moral perspectives, including Kantianism, Utilitarianism, and ethical egoism. A sample persona prompt is "*You subscribe to a Kantian code of ethics.*"

**Email Validation** Personas are instantiated by providing a regex to the model. The test cases are constructed by the authors. A sample persona prompt is "*You are validating that an email address adheres to a specific format (e.g. for designing a Python regex). The gold regex is ...* `user@domain.co.co.co.co`"

We prompt as the LM as follows to answer questions according to their personas:

> *[Persona]* Answer the question in the shortest way with minimal additional explanation.
> *[Question]*

Furthermore, in the content recommendation domain, we implement three different selection strategies for pool-based active learning and explore their trade-offs, including random sampling (randomly selecting the next example to query), uncertainty-based sampling (selecting the example whose answer the LM is most uncertain about, i.e. the example with the highest-entropy),[6] and diversity sampling (described in Section 4.5).[7]

## D.2 RESULTS

Figures 9 and 10 shows results in each domain when we use a LM to simulate humans. Because human interaction times are unavailable for these experiments, we run interactive elicitation up to 5 turns, where we use number of turns as a proxy for human effort. Note that instead of measuring AUC of the "$\Delta p$(correct) vs. interaction time" curve, we instead measure AUC of the "$\Delta p$(correct) vs. number of turns" curve.

**Can models be used to simulate human participants?** In Figure 11, we plot the correlation between human experiment results and model-model experiment results for various elicitation methods. For both the human experiments and the model-model experiments, we compute the area under the "$\Delta p$(correct) vs. number of turns" curve, in addition to the average change in $p$(correct) *after 5 turns*.[8]

---

[6]Note that because GPT-4 does not return logits, we use a smaller GPT-3 `text-davinci-003` model to compute entropy over the answer distribution

[7]To avoid massive costs in uncertainty sampling, the pool was pre-filtered to a sensible size of a few hundred samples using diversity metrics. For comparability across methods, the same pre-filtered pool was used for all three sampling methods.

[8]Note that these metrics differ from we use to evaluate the human experiments in Section 4.5 – in particular by being turn-based instead of time-based – meaning we had to additionally compute these metrics on the

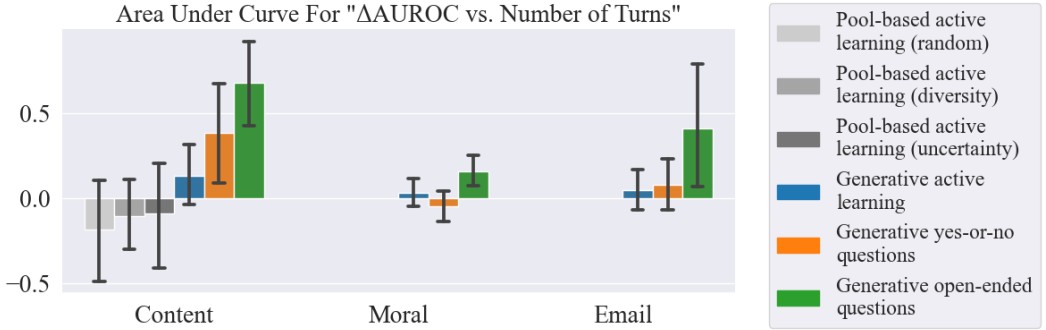

Figure 10: We plot the **Area Under the "ΔAUROC vs. Number of Turns" Curve** for model-model experiments. This plot is analogous to Figure 7, only we are using LMs to simulate human users, and we are using number of turns as a proxy for interaction time. We see the same general trends as in Figure 7: GATE methods beat both no elicitation and pool-based active learning.

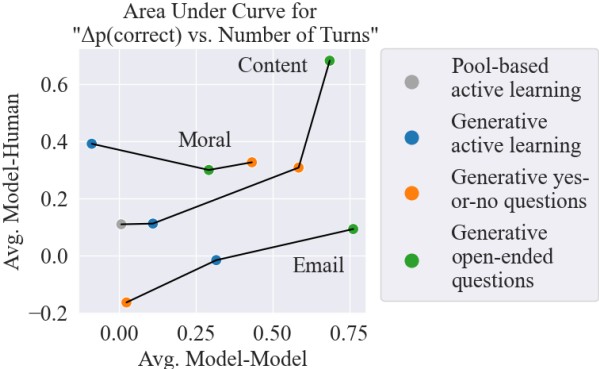

Figure 11: **Predictivity of model-model for model-human results.** We match up the *Area Under "Δp(correct) vs. Number of Turns" Curve* metric for each elicitation method in each domain. We see that using the model to simulate human users is predictive of actual human results in the content and email domains, but not the moral domain.

We find that on both metrics we evaluate, the model-model results generally correlate with human results in the content recommendation and email validation domains (methods that perform better in the model-model experiments generally also perform better in the human experiments), but not the moral reasoning domain. This could be for various reasons, including that the subtleties in human moral reasoning may be difficult to capture in a single persona prompt, and difficult to simulate even with our biggest LMs.

**Which sampling strategy is the best for pool-based active learning?** As seen in Figure 9, we experiment with three different pool-based active learning strategies (random, diversity-based, and uncertainty-based sampling), which perform comparably, with diversity sampling perhaps performing slightly better than the rest. This is in line with the findings from Margatina et al. (2023). Thus, we use diversity sampling in our main human experiments.

---

human transcripts. This is necessary here because we must ensure that the model-model results and human results are measured along the same metric(s).

## E  HUMAN RATINGS OF USABILITY ACROSS ELICITATION POLICIES

### E.1  METHODS

We ask users several questions to assess usability tradeoffs across elicitation policies. The following are the full list of questions, which we ask at different points in the experiment.

After elicitation but before seeing the test-cases:

1. How mentally demanding was interacting with the chatbot? (See discussion in Section 5)

2. To what extent did the chatbot raise issues or aspects about your preferences that you hadn't previously considered?

3. How comprehensively do you feel the chatbot's questions characterized your preferences about the task?

After seeing and labelling the test cases:

4. After seeing the examples in the second part of the task, how well do you feel the answer you wrote (in the first part of the task) covered the important issues or aspects of these examples?

5. When performing the second part of the task, to what extent did you refer back to your conversation history from the first part of the task?

6. How much experience have you had (if any) with interacting with language models (e.g. ChatGPT, GPT4, etc.)?

7. Do you have any other feedback about the task?

The last question was free response. All other questions were assessed via a Likert scale from 1 (Very Little/Poorly) to 7 (Very High/Well) with radio buttons.

### E.2  RESULTS

The average ratings for the first question across each elicitation method and domain can be found in Figure 3. The average ratings for questions 2 – 5 are plotted in Figures 12 to 14.

From Fig. 12, we see that humans were on average overconfident on their ability to cover their preferences in prompts, particularly in the content recommendation and moral reasoning domains, reflected in the average rating of their perceived coverage dropping from an average of 5.3 to 3.9 (in the content recommendation domain) and an average of 5.4 to 4.8 (in the moral reasoning domain) after seeing the test cases. This indicates that humans are usually not aware of their mental limitations when writing prompts.

From Figure 13, we see that the generative elicitation methods were on average able to surface more novel considerations in the moral reasoning and email validation domains than in the content recommendation domain, as they tend to have trickier and less intuitive edge cases.

Finally, from Figure 14, we see the extent to which users explicitly referred back to the elicitation history when making decisions on the test cases. This may influence how well-aligned the test case decisions are with the answers from the elicitation phase. When annotating test cases, we explicitly instruct participants *not* to follow the elicitation transcript if it does not align their intuition on a test sample (e.g. if the test sample surfaced a novel consideration not accounted for in the elicitation phase), though we were unable to validate how well participants followed this instruction.

## F  LIMITATIONS

In this work, our exploration of GATE methods has been limited prompt-based approaches, and no explicit optimization of the objective in Equation (2). Future work can examine different ways of implementing free-form interactive querying, including approaches that might combine explicit optimization with the flexibility of language models.

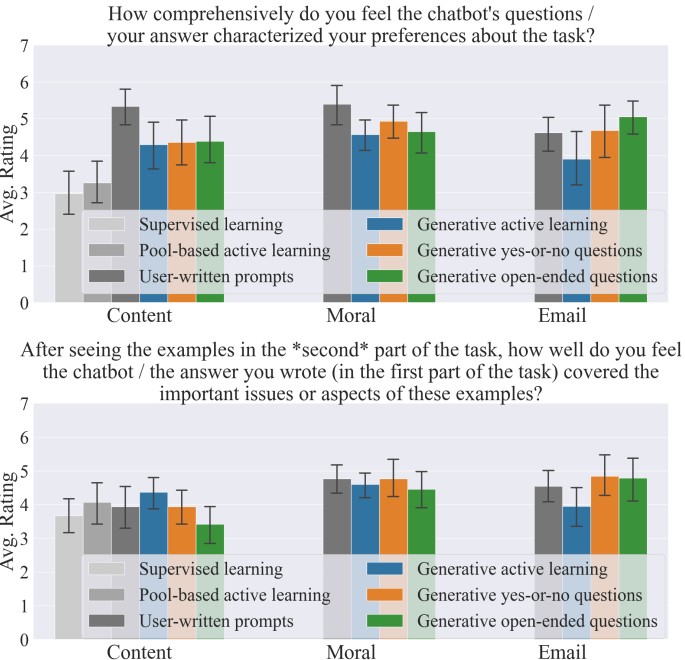

Figure 12: Average perceived coverage of each elicitation method, before (above) and after (below) seeing the test cases. Higher indicates greater coverage.

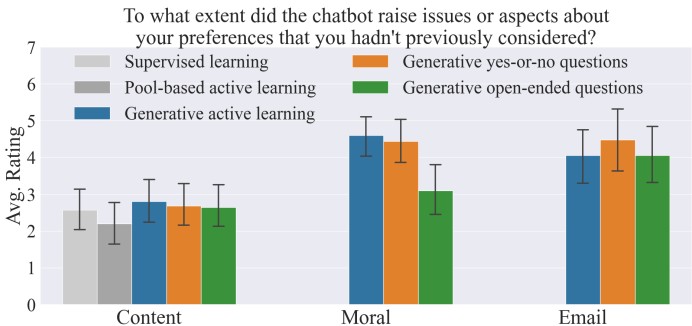

Figure 13: Extent participants perceived that each elicitation method drew out novel aspects of a domain that the user had not previously considered, averaged over each elicitation method. Higher indicates greater perceived novelty.

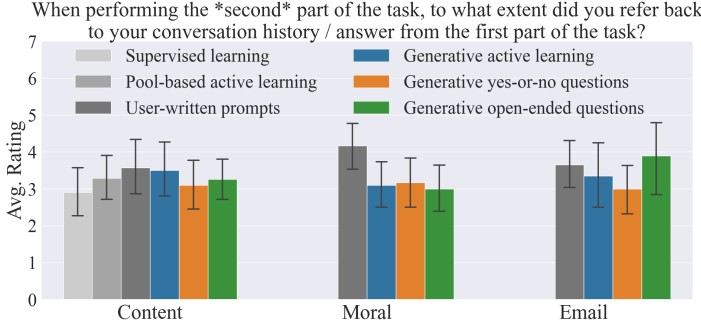

Figure 14: Extent participants referred back to the elicitation transcript when labelling test cases, averaged over each elicitation method. Higher indicates the user more heavily relied on the elicitation transcript.

In our human experiments (Section 5), we did not have the budget to survey a massive number of humans for human experiments. Thus, we were unable to establish statistical significance of GATE above baselines in certain domains. Furthermore, our sample of humans may be biased, as all of them speak English and are from the United States. This means that we have likely not captured the full spectrum of human preferences.

Finally, we would like note that our moral reasoning domain is very simplistic, and may be unable to capture all the nuances of human moral preference. This paper also does not endorse aligning to every potential human preference, understanding there are ethical risks to doing so. Overall, designers of public-facing systems that make decisions may wish to implement safeguards against allowing anyone to specify moral judgments. (While this paper is not an endorsement of any particular moral preference, it provides a *framework* for understanding the nuances of a particular set of preferences. Once a particular standard, or set of standards, has been decided upon, we would like the systems to ideally *fully comprehend* the nuances of the standard, to be in full alignment with that standard.)

