# OpenReview forum: "Eliciting Human Preferences with Language Models"
_ICLR.cc/2024/Conference — Submitted to ICLR 2024_

### Official Review · Reviewer_j3g4 · 2023-10-28

**Soundness:** 1 poor
**Presentation:** 3 good
**Contribution:** 3 good
**Rating:** 5
**Confidence:** 3

**Summary:**

This paper formalizes the task elicitation problem, under which existing learning paradigms are applicable. The authors also introduce GATE, a learning framework that uses language models to interact with users and elicit their preferences. The authors compare GATE to multiple other elicitation methods across three tasks and find that their proposed approach is comparable or sometimes better than baselines but at comparable or less mental demand for users.

**Strengths:**

- The paper introduces a nice framing to think about task elicitation and proposes a simple, but potentially effective method leveraging language models.
- I appreciated that the authors conducted a user study to evaluate users on multiple tasks, rather than simply using language models as a proxy for human users, and considered a number of relevant real-world metrics (e.g., mental load).

**Weaknesses:**

- While the paper introduces three different domains, two of the three baselines were not available for two of the three tasks. Since GATE shows promising results for content recommendations, it is hard to assess the validity and generalizability of the results without similar comparisons to baseline elicitation approaches for the other two tasks.
- It is unclear the extent to which the results found in this paper hinge on the choice of GPT-4 as both the LM that is used to elicit user preferences and the model that is used to predict user preferences. For the latter, prior work tends to train a reward model on the human data, rather than use the collected data as prompt input.
- It seems like elicitation using generative open-ended questions is the best of the three generative variants studied in this work (according to Figure 2), however, it also seems to be the most mentally demanding for the users (according to Figure 3) and seemed to be comparable to baselines. It would be nice to see a proposed modification to that approach that would maintain the *same* score in Figure 2 but would decrease the reported mental load, particularly given that the generative approaches that were tried in this work were all relatively simple (as stated by the authors themselves).

**Questions:**

Other clarification questions
- How is the framework in Section 2 connected to Section 3? The two sections felt a bit disconnected.
- Incorrect reference in Area under the p(correct)-time curve paragraph?
- In Figure 2, what does the * signify?
- What do 6/10 and 7/10 settings refer to?
- Does the term “time limit” refer to the 5-minute interaction period?

---

> ### Author Response · Authors · 2023-11-16
>
> **Baselines**
>
> Thanks for raising this point, as we could clarify better in the main text. The two primary baselines we consider are 1) No interaction (this is 0 on our graphs) and 2) Detailed prompts, where we have the participant write out their preferences. For the content preferences domain only, we also have access to a dataset of example news articles. We can use this to present two additional baselines: supervised learning and active learning. Unfortunately, in many domains, we do not have access to a large dataset of examples from the target domain; for example, there is no “natural distribution” over moral dilemmas or invalid email addresses. Thus, our experiments attempt to capture the performance of GATE in both settings.
>
> In fact we see that as an advantage of our method: lots of domains do not have large, pre-existing corpuses of data which allow for supervised learning and/or active learning. Our method opens the possibility of learning preferences in those domains without pre-collecting a labeled dataset.
>
> **Why not train a reward model?**
>
> We agree that reward models are a useful way to model preferences. Unfortunately, reward models require a large amount of data; for example, [1] collected a dataset of over 64,000 examples. By contrast, in our setting we consider interactions with a human that last less than 5 minutes long, which would leave us with far too little data to train a reward model. Furthermore, another advantage of using language models is their ability to comprehend free-form text data, rather than needing to learn from examples. As our transcripts are mostly free-form text (particularly in the two generative questions cases), it is impossible to train a reward model with traditional supervised learning.
>
> **Mediating generative open-ended questions**
>
> Broadly speaking, there is a tradeoff between interaction time/effort and the degree to which an approach can elicit human preferences. As the reviewer has noted, open-ended questions tend to be on one end of the tradeoff, requiring slightly higher mental demand (although equivalent mental demand to 2 of the 3 baselines), but being the best performing out of the three. Yes-or-no questions, while worse than open-ended questions at eliciting preferences, still beats out all baseline approaches, while being much equivalent or less mentally demanding to all three baselines. We agree that it would be great if there were some approach that could push the Pareto frontier, which we leave to future work.
>
> **How is the framework in Section 2 connected to Section 3?**
>
> In short, the framework of Section 2 motivates a gap in the literature which we fill in Section 3.
>
> Section 2 describes how existing learning paradigms can be described as task elicitation methods along two axes. First, methods can elicit information passively (as in supervised learning) or actively (as in active learning). Next, they can elicit information with examples (again as in traditional supervised learning) or by rich free-form inputs (such as prompting). However, no present method actively elicits free-form inputs. Section 3 describes our proposal of GATE, a framework for actively eliciting free-form task specifications, and we describe how we operationalize this framework for investigation in our paper. Section 4 then describes the concrete experiments we do given this operationalization.
>
> **In Figure 2, what does the * signify?**
>
> The asterisk signifies statistical significance at the 0.05 level. We have added this information to the caption in our newest revision.
>
> **What do 6/10 and 7/10 settings refer to?**
>
> Great catch – There were actually only 9 settings and GATE improves in 7 of them. Specifically, we examined the following 9 (domain, elicitation method) pairs from Figure 2 and counted in how many settings that GATE improves over user-written prompts, ignoring significance:
> (Content domain, generative open-ended questions): Yes
> (Content domain, generative yes-or-no questions): Yes
> (Content domain, generative active learning): No
> (Moral domain, generative open-ended questions): No
> (Moral domain, generative yes-or-no questions): Yes
> (Moral domain, generative active learning): Yes
> (Email domain, generative open-ended questions): Yes
> (Email domain, generative yes-or-no questions): Yes
> (Email domain, generative active learning): Yes
>
> 6/10 referred to a metric we had in a previous version of the paper, apologies for the confusion. We have taken this out in our newest revision and fixed the denominator to 7/9.
>
>
> **Does the term “time limit” refer to the 5-minute interaction period?**
>
> Yes, we have edited this in our newest revision to clarify.
>
>
> [1] https://arxiv.org/pdf/2009.01325.pdf

---

> ### Author Response · Authors · 2023-11-20
>
> Dear Reviewer j3g4,
>
> Thank you for your review and feedback. As the discussion period is coming to a close, please let us know if our response has adequately addressed your concerns, or if you have any remaining questions and concerns. If not, we would appreciate if you could raise your score. We appreciate your hard work!

---

> > ### Comment · Reviewer_j3g4 · 2023-11-20
> > **Follow-up question**
> >
> > Thank you authors for clarifying the scope of the paper. Could the authors discuss "the extent to which the results found in this paper hinge on the choice of GPT-4 as the LM that is used to elicit user preferences"?

---

> > > ### Author Response · Authors · 2023-11-21
> > >
> > > Great question -- we played around with prompting other models (e.g. Claude 2, Llama 2) to generate questions in our domains. While we didn’t have time to run a full study of these models, we were able to confirm that these models *are* able to generate reasonable-looking considerations similar to GPT-4.
> > >
> > > For example, Claude 2 generates the following questions for moral reasoning:
> > >
> > > > If a poor child steals bread from a large corporate grocery store chain to feed their hungry family, do you see that as ethical? Why or why not?
> > >
> > > > If a person broke into a private bakery at night and stole a loaf of bread without getting caught, would that be justified in your view?
> > >
> > > > What if the baker was struggling financially and the stolen bread represented a significant monetary loss? Would that change your opinion at all?
> > >
> > > > Imagine two children, one rich and one poor, both steal the same loaf of bread as a kind of game or dare. Is there any moral distinction in those two cases?
> > >
> > > > What if alternative sources of free or reduced-cost food were available (food banks, shelters, etc) but would require significant effort for the person to obtain? Would that change the ethical calculus around stealing?
> > >
> > > > If a grocery store regularly threw out loaves of bread at the end of the day rather than donating them, would taking that discarded bread still be considered stealing?
> > >
> > > the following questions for content recommendation:
> > >
> > > > What types of topics and subject areas are you most interested in reading about online? This can help me understand their interests.
> > >
> > > > Do you prefer articles that are short quick reads or longer in-depth analysis? This reveals preferences on article length.
> > >
> > > > When it comes to tone and style, do you like formal academic writing or more casual conversational pieces? This indicates preferences on tone.
> > >
> > > > What types of writing or storytelling do you find most engaging - investigative pieces, personal narratives, how-tos, etc? This points to preferences for different narrative formats.
> > >
> > > > How do you like articles formatted? Do you prefer text and images, audio accompanyments, infographics, etc? This provides insight on preferences for multimedia components.
> > >
> > > > Which websites or publications tend to have articles you enjoy reading? Are there any you actively avoid? This gives me examples to understand what's aligned or not aligned with their tastes.
> > >
> > > > How do you discover new articles to read online? Do you browse certain sites, use social media, or get suggestions from friends? Understanding discovery patterns can help tailor recommendations.
> > >
> > > and the following questions for email validation
> > >
> > > > What level of email address validation do you prefer - basic syntax checks or more advanced structure enforcement? For example, do you require the @ symbol and general user@domain format only, or do you have additional criteria like requiring a top level domain (".com") and at least 1 period in the domain part?
> > >
> > > > Should all types of characters be allowed in the user or domain parts as long as it meets general x@y.z formatting? Or should only alphanumeric characters and certain punctuation like periods, hyphens or underscores be permitted?
> > >
> > > > Do you want to accommodate both upper and lowercase letters or enforce lowercase only in the address?
> > >
> > > > Do you have a length requirement like maximum of 64 characters total? Or restrictions on user or domain part lengths individually?
> > >
> > > > For the domain, should common top level domains like .com, .net be assumed valid without actually pinging a server or do you want to implement a reference list of allowed TLDs based on known domain extensions?
> > >
> > > > Are there any special formatting rules you want enforced like disallowing consecutive periods, requiring usernames that are at least 2 characters, etc?

---

> > > > ### Comment · Reviewer_j3g4 · 2023-11-21
> > > >
> > > > Thanks to the authors for their continued efforts to improve this submission. I have raised my score from a 3 to a 5 as a result of the author's clarifications. I still believe that this work has limitations that I hope the authors could address in the future in terms of more precisely defining the scope of when GATE should be used and a thorough evaluation in comparison to relevant baselines in both settings that the authors mentioned in their first response.

---

### Official Review · Reviewer_d4UR · 2023-11-01

**Soundness:** 3 good
**Presentation:** 3 good
**Contribution:** 3 good
**Rating:** 6
**Confidence:** 4

**Summary:**

The paper proposes an approach for an interactive recommeder system on the basis of LLMs, here GPT4. The authors suggest three methods- "Generative active learning", "Generating yes-or-no questions, "Generating open-ended questions" - for which the propose respective LLM prompts. The evaluation is conducted for three domains, namely content recommendation, moral reasoning, email verification by means of a user study including in total 388 participants. The system is compared to standard supervised learning trained on Microsoft News Dataset (only for content rec.), a pool-based active learner and user-made prompts for preferences. The results show that the three proposed methods are either on par or even outperforming the baseline approaches for the domains. In addition, the felt mental demand of the participants was mostly lower for the proposed methods, with open-ended questions being the most demanding among the three proposed approaches.

**Strengths:**

The research direction is fruitful for LLMs, as their capability to continuously, autonomously interact with end-users is crucial. To this end, the three proposed methods are sensible. It is therefore insighful and value adding to evaluate the performance of the respective prompts.

The paper clarity is quite high, showing detailled evaluation results, method descriptions and related works.

The empirical results are promising, showing that LLMs can guide the user towards the correct goal. In addition, the chosen baselines are quite well-chosen, making the results significant. The conducted user study is well-structured and a important part for the contribution of the paper. Here, the results are very insightful and can support future prompt design or fine-tunings. It is informative and very relevant to see that the different proposed prompts have different advantages in the respective domains and felt mental loads.

**Weaknesses:**

It is unclear where other known preference elicitation approaches such as pairwise comparisons / choice-based preference elicitation / ... fit into the stated related work. They might be situated in example-based/interactive, but especially there are other studied interaction mechanisms in this field which might have overlaps to free-form. It should be part of the related work coverage.

Part of the investigated methods (generative active learning and generating yes-or-no questions) might be too rigid, as a interests/preferences might not be black and white. I understand that the user can give any response to any of the methods, thereby giving more refined answers than yes/no, but handling arbitrary relative answers might become difficult to handle. The paper does not analyse the availability of such a problem / rule it out based on the results of the conducted user study. Here, ablations of possible user answers or prompt changes might be insightful

As mentioned in the reproducibility section, the authors used closed-source GPT4 for their experiments, which makes exact replication difficult.

**Questions:**

Did you test implications of answer diversity on the success of your methods? Taking yes/no questions, a user might still answer with any fuzzy statement which might be quite ambiguous. It would be intersting to know if limiting the answers to potentially some selected "in-between" categories would help or not. Along this, did you analyse how often a user answered with such an ambiguous statement?

Would it be possible to combine the proposed prompts to improve performance / reduce mental load? Did you experiment with such prompts? The open-ended version could, of course, do so by design - did you conduct analyses if this happens?

---

> ### Author Response · Authors · 2023-11-16
>
> We thank the reviewer for their review!
>
> **Other preference elicitation formats**
>
> This is a great point! This is indeed relevant literature which we will discuss in related work. We believe that GATE is complementary to pairwise comparisons / choice-based elicitation, and we can definitely do them within the GATE framework (i.e. prompt the LM to generate pairwise queries, choices to select among), which would be a good topic for future work. We don't mean to claim that this paper has exhaustively characterized the space of possible GATE policies.
>
> **Some answer types are less flexible**
>
> We agree—generative active learning and generative yes/no questions both assume a binary set of responses.  This might be limiting in real-world settings where the true answer might be “it depends,” or something more subtle. Interestingly, however, we do see some evidence that participants use the open-endedness of the chat interface to provide more details in these settings, rather than always replying with a single “yes” or “no.” And in general we believe that the advantage of our work is that it opens up the possibility for open-ended dialogue and answers that can more fully capture the space of nebulous human preferences, as we explore in the open-ended response settings we study.
>
> **Choice of models**
>
> We thank the reviewer for this note. While we consider the GATE framework in our paper to be the main contribution of the work, we agree that closed-source models are limited. We hope to revisit these experiments with open-source models in future work, especially as more able models are released.
>
> **Answer diversity vs. success of GATE**
>
> This is a great question! We did a preliminary investigation of whether we could correlating the **% of answers that aren’t “yes” or “no”** in a transcript (for the “yes-or-no questions” elicitation setting) vs. the **delta p(correct)** after the transcript. Unfortunately, we could not find a strong correlation in our preliminary investigation (p-values $0.4$, $0.5$, $0.9$ respectively for the email, moral, and content domains). Note that writing an explanation takes additional time, so transcripts with more explanations tend to also have less turns overall. This preliminary result suggests that directly answering the questions asked by GATE may be more useful than whatever explanation a user may decide to provide. However, more fine-grained analyses may be necessary to break down the nature of the additional explanation / fuzzy answer, and the direct contribution of that answer to the accuracy.
>
> **Combine proposed prompts to improve performance & reduce mental load**
>
> This is a good question. We believe that combining question types would be an interesting direction for future work, and could be better able to navigate tradeoffs between performance vs. mental load. We will add this discussion to future versions of our paper. We also found that the open-ended case does indeed sometimes ask yes-or-no questions.

---

> > ### Author Response · Authors · 2023-11-20
> >
> > Dear Reviewer d4UR,
> >
> > Thank you for your review and feedback. As the discussion period is coming to a close, please let us know if our response has adequately addressed your concerns, or if you have any remaining questions and concerns. We appreciate your hard work!

---

> > > ### Comment · Reviewer_d4UR · 2023-11-21
> > > **Thank your for your clarifications!**
> > >
> > > Thanks for your answers! Yes, I think it would be important to add the mentioned works to the related work section, as they can be seen as alternative or extension to active learning in your problem setting.
> > >
> > > I believe your conducted evaluation nicely analyses/discusses the relative merits of the selected approaches. I was additinally wondering how to interpret the results "absolutely" - would a user with reached AUC results (for your approaches) be already using the system or rather search residing to manual search?

---

> > > > ### Author Response · Authors · 2023-11-21
> > > >
> > > > Dear reviewer d4UR,
> > > >
> > > > We agree these works are highly relevant and will definitely add them to the related work section.
> > > >
> > > > If we are interpreting your question correctly, you're asking whether users would use GATE rather than manual methods. We believe in our domains that the natural "manual baseline" is prompting, a standard approach for interacting with modern LMs. We provide comparisons to prompting in all three domains and find that our method generally outperforms prompting. Furthermore, we'd also like to clarify that we're not claiming to have "solved" any of these tasks in an absolute sense---we're doing better than the manual baseline but there's still a lot of headroom

---

### Official Review · Reviewer_Nhj6 · 2023-11-01

**Soundness:** 3 good
**Presentation:** 4 excellent
**Contribution:** 3 good
**Rating:** 6
**Confidence:** 2

**Summary:**

The paper addresses the issue of training LLMs to perform complex task such as personalized website recommendations via interaction with the users in an active learning mode.

**Strengths:**

+ comprehensive and well written paper
+ introduction of a novel method GATE for generative open-ended questions and model improvement with active learning
+ ethical considerations and limitations are well thought
+ code availability

**Weaknesses:**

- the testing is well-thought but quite limited. Such models require an extensive testing to ensure they are not overfitting, especially when for systems like this one which may eventually foster polarization as an effect of extreme self-centred recommendations.
- the concept of morality is underdefined. What is the notion of morality employed here? There is quite some literature on the moral foundations theory and their detection from text which should be used as a benchmark for the model

**Questions:**

- the testing is well-thought but quite limited. Such models require an extensive testing to ensure they are not overfitting, especially when for systems like this one which may eventually foster polarization as an effect of extreme self-centred recommendations.
- the concept of morality is underdefined. What is the notion of morality employed here? There is quite some literature on the moral foundations theory and their detection from text which should be used as a benchmark for the model

---

> ### Author Response · Authors · 2023-11-16
>
> We thank the reviewer for their review!
>
> **Overfitting and polarization**
>
> We agree it is crucial to prevent overfitting to people's preferences. Our work makes efforts to guard against two potential sources of overfitting. First, we guard against statistical overfitting of our classifiers by ensuring that models are always evaluated on how well they can predict preferences on held-out data from another distribution. The strong performance of our models in this setting indicates they are not overfitting to the training data. Second, we check that the model does not cause the humans to “overfit” by strengthening or modifying their own preferences (e.g. via echo chamber effects). Notably, Figure 3, right, demonstrates that people's preferences do not change after using GATE. That said, we agree that longer-term use of personalized recommender systems warrants caution and further study, and have included discussion of this in our ethical considerations section.
>
> **Morality**
>
> We completely agree that there is no single objective framework for moral reasoning. The point of these experiments in our paper was to see whether GATE could enable a language model to elicit information about an individual’s subjective, personal moral preferences. Interestingly, we do find that many of the questions asked by the language model align with the axes explored by moral foundations theory (e.g. exploring the tension between care and fairness), and have added discussion and citations to that effect in the newest revision.

---

> > ### Author Response · Authors · 2023-11-20
> >
> > Dear Reviewer Nhj6,
> >
> > Thank you for your review and feedback. As the discussion period is coming to a close, please let us know if our response has adequately addressed your concerns, or if you have any remaining questions and concerns. We appreciate your hard work!

---

### Official Review · Reviewer_djQq · 2023-11-07

**Soundness:** 2 fair
**Presentation:** 3 good
**Contribution:** 2 fair
**Rating:** 5
**Confidence:** 4

**Summary:**

The authors propose using LLMs to elicit or derive from the user what their preferences are based on a new method called GATE. Here the LLM asks the user either yes or no, or open ended questions to find out what their preferences are for recommendations. The research experiment shows improvement in novel recommendation considerations.

**Strengths:**

I am afraid I cant see any strength.

**Weaknesses:**

The authors of this research project have not considered the ethical aspects of having LLMs ( they already have their own bias and hallucination issues) probe humans to spill their preferences. Interacting with a chatbot an/or  AI systems have shown to be psychologically endearing to humans and yet the researchers seem to be intent on making recommendations more accurate than the population's psychological impact.
No in depth assessment of conversation based interactions of this nature and their strength of cognitive processes in gathering accruate recommendations outside of this field was provided.

**Questions:**

Please provide any ethical assessments done during or prior to IRB approval of how such a system can impact humans over the long term use of a GATE based recommendation system? Please provide more depth to this " A fundamental challenge across many fields is how to obtain information about people’s nebulous
thoughts, preferences, and goals. In psychology and cognitive science, protocol analysis describes
methods for how to obtaining and analyze verbal reports from subjects about cognitive processes
including via think-aloud protocols (Ericsson & Simon, 1980; Ericsson, 2017)"

---

> ### Author Response · Authors · 2023-11-16
>
> We thank the reviewer for their review!
>
> **Long-term use of GATE**
>
> As the reviewer noted, our research received IRB approval for the experiments we carried out in our paper. These experiments involved short-term interactions with GATE; specifically, participants interacted with the model for five minutes. However, we did consider whether interacting with a language model would have an impact on user preferences (Figure 3, right). There, we found that participants did not have significantly different preferences before vs after interacting with the model; that is, using GATE did not materially alter the user’s preferences. We agree that the long-term use of language model assistants is worth caution and further study. In our ethical considerations section we will expand on our discussion of “thin slicing” and other harms to include discussion of potential attachments people may form with language models and how to avoid dependence and over-sharing. We will also caution against widespread long-term use of LM companions prior to these risks being better understood.
>
> **Discussion of work in other fields**
>
> We are happy to elaborate on these sentences. Many fields are concerned with understanding people’s cognitive processes, inner states, or personal preferences. For example, in cognitive psychology one might attempt to understand what strategies people use when attempting to solve math problems, and in user experience research one might attempt to understand how users understand an interface. However, these inner experiences are often hard to verbalize because people are not always skilled at introspecting and discussing what factors guide their preferences and actions. Thus, many fields study how to better elicit and measure these inner cognitive states, including by interviewing and asking questions of people, as we study in our work

---

> ### Author Response · Authors · 2023-11-20
>
> Dear Reviewer djQq,
>
> Thank you for your review and feedback. As the discussion period is coming to a close, please let us know if our response has adequately addressed your concerns, or if you have any remaining questions and concerns. If not, we would appreciate if you could raise your score. We appreciate your hard work!

---

> > ### Comment · Reviewer_djQq · 2023-11-20
> > **Followup**
> >
> > Thanks for responding to the review. Can you please elaborate or share more references that you have found in your research about conversation based interactions in any field of study and their validity in extracting accurate preference knowledge other than what is listed in the paper? It appears to me that your entire premise of the GATE stands on that assumption that such interactions are worthwhile. From my awareness, the more you ask, the more you probe, the less useful the information is. Based on cognitive and market studies. But you may have found evidence to the contrary which I don't see listed in your references - do send us one or two such studies on which your foundation is built on.

---

> > > ### Author Response · Authors · 2023-11-21
> > >
> > > Dear reviewer djQq,
> > >
> > > We are happy to share further references. In survey research, structured interviews are often used to reveal people's opinions (aka preferences) through conversation. Though these interviews utilize mostly fixed questions, open-ended questions may also be included. While it's true that there are cases in which decisions reveal underlying preferences inconsistent with overt reports, these are the interesting second order corrections not the main effect. [1,2,3,4,5]
> > >
> > > The review mentions "the more you ask, the more you probe, the less useful the information is based on cognitive and market studies". Can you provide references to the specific studies you're thinking of?
> > >
> > > [1] Wright, P. M., Lichtenfels, P. A., & Pursell, E. D. (1989). The structured interview: Additional studies and a meta‐analysis. Journal of occupational psychology, 62(3), 191-199. https://bpspsychub.onlinelibrary.wiley.com/doi/abs/10.1111/j.2044-8325.1989.tb00491.x
> > >
> > > [2] Kallio, H., Pietilä, A. M., Johnson, M., & Kangasniemi, M. (2016). Systematic methodological review: developing a framework for a qualitative semi‐structured interview guide. Journal of advanced nursing, 72(12), 2954-2965. https://onlinelibrary.wiley.com/doi/full/10.1111/jan.13031
> > >
> > > [3] Barriball, K. L., & While, A. (1994). Collecting data using a semi-structured interview: a discussion paper. Journal of Advanced Nursing-Institutional Subscription, 19(2), 328-335.
> > >
> > > [4] Weisberg, H., Krosnick, J. A., & Bowen, B. (1996). Introduction to survey research, polling, and data analysis. Thousand Oaks, CA: Sage. https://psycnet.apa.org/record/1997-97082-000
> > >
> > > [5] Ponto J. Understanding and Evaluating Survey Research. J Adv Pract Oncol. 2015 Mar-Apr;6(2):168-71. Epub 2015 Mar 1. PMID: 26649250; PMCID: PMC4601897. https://www.ncbi.nlm.nih.gov/pmc/articles/PMC4601897/

---

### Meta-Review · Area_Chair_Abqb · 2023-12-24

**Metareview:**

This paper looks at the problem of task specification for language models, that is, whether you get LMs to generate the task, then ask the user for feedback. They propose the generative active task elicitation (GATE) framework that show promising results. Compared to baseline methods (supervised learning using user-labeled data, no elicitation, and user-provided task descriptions), the GATE method is sometimes more accurate and causes less cognitive burden to the users.

Except for one of the reviewers who was more concerned with the long-term ethics of user preference elicitation, reviewers are generally positive about the problem and the contributions. However, there are some loose ends that should be addressed before publication. They include

- How would the GATE framework do with open models such as LLaMa 2? Authors do provide some examples in the rebuttal but would need more time to do an in-depth analysis.
- What are some other types of questions to elicit user preferences? How would they work in the GATE framework?
- What is the scope of the effectiveness of GATE? What types of tasks and what types of elicitation questions would work best?

With answers to these (and perhaps others, please see reviews for more detail), this paper would make a really meaningful and significant contribution to the important topic of learning paradigms for LLMs. I strongly encourage the authors to resubmit to a future venue with these revisions.

**Justification For Why Not Higher Score:**

The three bullet points above (using an open LLM, exploring elicitation question types, defining the scope) should be addressed for a higher score.

**Justification For Why Not Lower Score:**

N/A

---

### Decision · Program_Chairs · 2024-01-16

Reject